# Periodontal Treatment Experience Associated with Oral Health-Related Quality of Life in Patients with Poor Glycemic Control in Type 2 Diabetes: A Case-Control Study

**DOI:** 10.3390/ijerph16204011

**Published:** 2019-10-19

**Authors:** Yuan-Jung Hsu, Kun-Der Lin, Jen-Hao Chen, Mei-Yueh Lee, Ying-Chu Lin, Feng-Chieh Yen, Hsiao-Ling Huang

**Affiliations:** 1School of Dentistry, College of Dental Medicine, Kaohsiung Medical University, Kaohsiung 80708, Taiwan; 2Department of Oral Hygiene, College of Dental Medicine, Kaohsiung Medical University, Kaohsiung 80708, Taiwan; 3Department of Medicine, College of Medicine, Kaohsiung Medical University, Kaohsiung 80708, Taiwan; 4Division of Endocrinology and Metabolism, Department of Internal Medicine, Kaohsiung Medical University Hospital, Kaohsiung 80708, Taiwan; 5Department of Internal Medicine, Kaohsiung Municipal Ta-Tung Hospital, Kaohsiung Medical University Hospital, Kaohsiung 80145, Taiwan; 6Department of Dentistry, Kaohsiung Municipal Hsiao-Kang Hospital, Kaohsiung 812, Taiwan; 7Division of Endocrinology and Metabolism, Kaohsiung Municipal Hsiao-Kang Hospital, Kaohsiun 812, Taiwan; 8Division of Endocrinology and Metabolism, Chi Mei Medical Center, Tainan 710, Taiwan

**Keywords:** Glycemic control, oral health-related quality of life, periodontal care behavior, periodontal treatment experience

## Abstract

Severe periodontitis is a risk factor for poor glycemic control. The appropriate medical treatment and plaque control of periodontitis positively affects blood-sugar control in diabetes patients. We aimed to identify the factors associated with glycemic control and examine the periodontal treatment (PT) experience and oral health-related quality of life (OHQoL) for patients with poor glycemic control in type 2 diabetes mellitus (T2DM). This multicenter case–control study recruited 242 patients with poor glycemic control and 198 patients with good glycemic control. We collected patients’ information through face-to-face interviews using a structured questionnaire. The Oral Health Impact Profile-14 (OHIP-14) was used to measure OHQoL. Based on PT status, the patients were classified into three groups: a non-periodontal disease group, a PT group, and a non-PT (NPT) group. Regression models were used to analyze the data. No interdental cleaning (adjusted odds ratio (aOR) = 1.78) and positive attitudes toward periodontal health (aOR = 1.11) were significantly more likely to be associated with poor glycemic control in patients with T2DM. The PT group had a significantly lower OHIP-14 score than the NPT group (6.05 vs. 9.02, *p* < 0.001), indicating a better OHQoL among patients with poorly controlled T2DM. However, the OHQoL did not differ significantly in patients with well-controlled T2DM between the PT and NPT groups. This suggested that diabetic patients with poor glycemic control must improve periodontal care practices and receive proper PT, if necessary, to improve their OHQoL.

## 1. Introduction

Diabetes is a systemic and irreversible disease that has become an increasingly crucial public health concern worldwide. The global prevalence of diabetes in 2017 was approximately 425 million (8.8%) and is expected to rise to 629 million by 2045 [1]. In Taiwan, approximately 1.96 million (11%) adults aged 20–79 years have diabetes [1]. In 2017, diabetes ranked fifth among the 10 leading causes of death; deaths caused by diabetes account for approximately 5.7% of all deaths [2]. Improper care can result in numerous complications such as cerebrovascular disease [3], cardiovascular disease [4], kidney disease [5], diabetic retinopathy [6], and peripheral neuropathy [7]; it may also ultimately lead to blindness, dialysis, and amputation. A history of severe hypoglycemia and hyperglycemia (“severe dysglycemia”) and the burden of diabetes complications are also independent risk factors for repeat hospitalizations [8]. The type 2 diabetes mellitus (T2DM) patients with a glycated hemoglobin level (HbA1c) ≤ 6.5% were more likely to have a good-excellent quality of life (QoL) as compared with those with an HbA1c level > 6.5% [9].

Possible mechanisms underlying the interaction between diabetes and periodontal disease have been investigated [10]. The oral complications of diabetes include xerostomia, dental caries, gingivitis, periodontal disease, and an increased tendency of oral infections [11], with periodontal disease being the most common oral disease among diabetes patients. Individuals with diabetes are at greater risk for chronic periodontitis and have more severe chronic periodontitis than individuals without diabetes. In the third US National Health and Nutrition Examination Survey, the incidence of periodontitis was twofold in patients with diabetes than in those without (17.3% vs. 9%); similarly, the incidence of diabetes was twofold in patients with periodontitis than in those without (12.5% vs. 6.3%) [12]. A Taiwanese study also found that 73.5% of the type II diabetes patients had periodontal disease (community periodontal index (CPI) ≥ 3), and there was a significant correlation between periodontal disease and HbA1c level (≥8.0%) [13]. 

Periodontal disease may directly affect the oral health-related quality of life (OHQoL), aspects of which include difficulty in speaking, halitosis, discomfort when eating, taste disorders, and general dissatisfaction with oral health [14]. A study assessed the oral health and health-related quality of life (HRQoL) of people with and without T2DM using a 36-item short-form survey; patients with T2DM and few remaining natural teeth received poor results in the physical functioning and role function-physical domains than did those with 20 or more teeth [15]. Another study investigating the HRQoL of elderly people with T2DM determined that a low HRQoL was correlated with poor oral health and a lack of dental care [16].

Clinical research has indicated that the appropriate medical treatment of periodontitis positively affects blood-sugar control in diabetes patients [17,18]. Grossi et al. [19] concluded that after a combination of dental treatments and antibiotics, there is a clear improvement in the periodontal condition and blood-sugar control. A study conducted with a randomized clinical periodontal intervention on diabetes patients suggested that after 18 months, the HbA1c in the experimental group had significantly improved compared with the control group [20]. Another randomized clinical test on type II diabetes patients also showed that the HbA1c in the experimental group had significantly improved in comparison with the control group that had not undergone periodontal treatment intervention [21]. Katagiri et al. [22] conducted a study in which the experimental group underwent periodontal treatment and the control group received only oral hygiene education. After following up over a period of six months, the HbA1c in the experimental group had significantly improved compared to the control group.

Severe periodontitis is a risk factor for poor glycemic control [23]. Patients with poor glycemic control have more severe bleeding, bone loss, and periodontal tissue destruction than those with good glycemic control [23,24]. Though periodontal disease can adversely affect the metabolic control of diabetes [25], oral health care is not incorporated in treatment planning for diabetes patients. The self-management of diabetes is essential to control their disease and to reduce the risks of associated disabilities. The prevention and management of periodontitis may be important for the successful management of T2DM. The appropriate periodontal treatment (PT) and plaque control of periodontitis positively affects periodontal condition and blood-sugar control in diabetes patients. Plaque control is divided into oral self-care, self-performed plaque removal at home, and professional mechanical plaque (and calculus) removal by a dentist. No literature has explored whether periodontal care behaviors may directly affect glycemic control status. Therefore, we conducted the present research focusing on patients with poor glycemic control in T2DM to identify associated factors related to PT and oral self-care behaviors. We also hypothesized a significant difference in the OHQoL between patients with poorly controlled T2DM who had received PT and those who did not receive PT. 

## 2. Materials and Methods

### 2.1. Design and Participants

This study was a multicenter case–control study. We recruited our participants from the endocrinology and metabolism divisions of Kaohsiung Medical University Hospital, Kaohsiung Municipal Hsiao-Kang Hospital, and Tainan Chi-Mei Medical Center in Taiwan from 2014 to 2015. Participants who (1) had type 1 diabetes mellitus, (2) routinely used antibiotics or bisphosphonates, (3) had a history of cancer, (4) were completely edentulous, and (5) had no data on glycated hemoglobin (HbA1c) levels within the previous 6 months were excluded from both groups of the study.

According to the definition of American Diabetes Association [26], the case patients in our study had T2DM with poor glycemic control, which means the HbA1c levels obtained from the electronic medical records system of the hospital for the previous 6 months were ≥7%. The controls were T2DM patients recruited from the same hospitals during the same period. The controls had good glycemic control, which means the HbA1c levels obtained from the electronic medical records system of the hospital for the previous six months were <7%. We calculated that 100 patients in each group would provide 80% power (two-sided type 1 error of 5%) to detect a 0.20 effect size for the level of OHQoL between the 2 groups. In total, 242 case patients and 198 controls were included in this study for further analyses.

### 2.2. Questionnaire Development and Measures

A structured questionnaire was developed to collect data in this study. The questionnaire was comprised of four parts. The first part was related to demographic characteristics. The second part included periodontal treatment experience and periodontal self-care behaviors. The third part included oral health-related knowledge and attitude toward periodontal health. The fourth part obtained information regarding the OHQoL. This questionnaire was based on established and validated questionnaires used in recent studies [27,28]. The questionnaire was reviewed by a panel of experts to assess its content and validity. Based on the results of content validity index, items were revised to enhance clarification and appropriateness. To ensure that the content was understood by our participants, the questionnaire was then piloted to thirty T2DM patients. 

#### 2.2.1. Demographic Characteristics 

The demographic characteristics encompassed nine items. The major characteristic variables included gender, age, the duration of diabetes, and education level. Other data, such as poor oral hygiene habits (drinking, betel nut chewing, and smoking) and perceived oral health status, were collected to understand the basic information of the participants.

#### 2.2.2. PT Experience

PT experience was assessed through two self-report questions: “Have you ever been diagnosed as having periodontal disease by dentists?” If the patients answered yes, the subsequent question was “Have you ever received PT from dentists?” The answers were used to determine the PT groups. The participants were classified into non-periodontal disease (NPD), non-PT (NPT), and PT groups.

#### 2.2.3. Periodontal Care Behaviors

Patients were asked questions regarding their tooth brushing time, tooth brushing method, routine dental visit, and interdental cleaning (interdental brushing or dental flossing). The brushing method was coded as 0 (other) or 1 (modified Bass brushing technique). Brushing duration was coded as 0 (<2 min) or 1 (≥2 min). Interdental cleaning behavior was assessed through the item “In the last half year, did you have a habit of using floss or interdental brush?” The response to this item was coded as 0 (never) or 1 (at least once daily). Routine dental visit was coded as 0 (no) or 1 (yes, having dental checkup or scaling in 6 months).

#### 2.2.4. Oral Health-Related Knowledge

A 7-item scale regarding oral health-related knowledge was referred by to Yuen [27]: “People with diabetes are more likely to have gum disease;” “People with diabetes are more likely to have infection in their mouth;” “Diabetes can make one’s teeth and gums worse;” “People with dry mouth are more likely to have a sore in the mouth;” “People with dry mouth are more likely to have tooth decay;” “If your gums bleed every time you brush your teeth, it’s an early sign of gum disease;” and “Gum disease can lead to loss of teeth.” Possible responses included true, false, or unknown, with possible scores ranging from 0 to 7; higher scores indicated a better degree of oral health-related knowledge. A 0.79 on the Kuder–Richardson reliability test was acceptable.

#### 2.2.5. Attitude toward Periodontal Health

The 7-item scale was employed to measure attitude toward periodontal health: “There is a relationship between periodontal disease and systemic health;” “Good periodontal health is important to overall health;” “Brushing and flossing daily are essential to maintaining periodontal health;” “People with poor periodontal health cannot control their glycemic index;” “Good glycemic control can reduce the severity of periodontal disease;” “The condition of our teeth can influence our dietary intake;” and “People with diabetes should care more about oral health.” The items were evaluated using a 5-point Likert scale with ratings from 1 (strongly disagree) to 5 (strongly agree), with possible scores ranging from 7 to 35. Cumulative scores were summed, with higher scores reflecting more positive attitudes toward periodontal health. In our pilot test, the Cronbach’s α for the scale was 0.76.

#### 2.2.6. OHQoL

The OHQoL was measured using the 14-item Taiwanese version of the Oral Health Impact Profile-14 (OHIP-14T) and was deemed to have high internal consistency, with a Cronbach’s α of 0.90 [28]. The OHIP-14T comprised seven domains: functional limitation, physical disability, physical pain, psychological discomfort, psychological disability, social disability, and handicap. The response format was delineated as follows: very often = 4, often = 3, occasionally = 2, rarely = 1, and never = 0. The scores ranged from 0 to 56; lower scores indicated a better OHQoL.

### 2.3. Data Collection Procedure

The patients’ HbA1c levels within the previous 6 months were checked prior to data collection. Physicians asked their patients if they were willing to participate in the research when these patients visited them during general clinic hours. Subsequently, a trained researcher explained the aim of the study to the participants and asked them to sign the informed consent form. Our questionnaire was used to collect data during a face-to-face interview conducted by a well-trained interviewer.

### 2.4. Statistical Analysis

This study explored the relationships among the variables using STATA version 13.0 (Stata Corp LP, College Station, Texas, USA). Logistic regression models were used to analyze variables related to glycemic control status and the relationships between other variables and periodontal care behaviors. The multivariate-adjusted means and differences (partial regression coefficients) were used to analyze the seven domains of OHIP-14T among the three PT groups stratified by glycemic control. We adjusted the results for potential confounders, namely age, gender, T2DM duration, education level, and routine dental visit.

### 2.5. Human Ethics

This study was approved by the Institutional Review Boards of Kaohsiung Medical University Hospital (KMUH-IRB-20130374) and Tainan Chi-Mei Medical Center (IRB-10307-004). We obtained informed consent forms from all participants.

## 3. Results

### 3.1. Distribution of Sociodemographic Characteristics 

The distribution of sociodemographic characteristics was examined in T2DM patients stratified by glycemic control status (Table 1). The proportion of patients with poor glycemic control who had a diabetes duration of >10 years was higher (30.4%) than that of controls (12.2%), with a statistically significant difference between both groups (*p* < 0.001). Furthermore, a significantly higher proportion of patients with poor glycemic control had a betel nut chewing habit (6.2%) compared with controls (2.0%); a similar result was obtained for smoking habit for the case (18.6%) and control (11.6%) groups. No significant differences were observed for other items between the two groups.

### 3.2. Glycemic Control Status Associated with Selected Variables in T2DM Patients

Table 2 lists the adjusted odds ratios (aOR) for the association of glycemic control status and selected variables in patients with T2DM. Patients with more positive attitudes toward periodontal health and without interdental cleaning behavior were 1.11 (95% confidence interval (CI) = 1.02, 1.21) and 1.76 (95% CI = 1.05, 2.96) times more likely to have poorly controlled diabetes, respectively.

### 3.3. Periodontal Treatment Experience and Periodontal Care Behaviors in T2DM Patients in Multivariate Regression Model

The results of multivariate logistic regression analyses for PT experience and selected variables related to periodontal care behaviors in patients with T2DM are presented in Table 3. In terms of PT experience, patients in the PT group were more likely to report brushing teeth with the Bass technique (aOR = 1.77, 95% CI = 1.09, 2.87) and having interdental cleaning (aOR = 2.47, 95% CI = 1.39, 7.39). Patients with a high education level (higher than technical school or college) were more likely to brush their teeth for two minutes or more (aOR = 2.00, 95% CI = 1.08, 3.70). After adjusting for potential confounders, the other characteristics deemed to be significantly related to interdental cleaning behavior of patients with T2DM were female gender (aOR = 1.82, 95% CI = 1.07, 3.10), an education level of high school (aOR = 1.92, 95% CI = 1.03, 3.60) and higher than technical school or college (aOR = 2.95, 95% CI = 1.42, 6.11), and positive attitudes toward periodontal health (aOR = 2.47, 95% CI = 1.39, 7.39). No other variables were significantly related to routine dental visits in T2DM patients.

### 3.4. The Levels of OHQoL in Different Periodontal Treatment Experience Groups Stratified by Glycemic Control Status

The levels of OHQoL of the three PT groups stratified by glycemic control are illustrated in Figure 1. Patients with well-controlled T2DM in the NPD group had a significantly lower OHIP-14T score than did those in the other groups. Though the difference was nonsignificant, the PT group had a lower OHIP-14T score than the NPT group (7.99 vs. 8.43). Patients with poorly controlled T2DM in the PT group had a significantly lower OHIP-14T score than did those in the NPT group (6.05 vs. 9.02). The OHIP-14T scores did not differ significantly between the NPD and PT groups.

### 3.5. OHIP-14 Domains in Periodontal Treatment Groups Stratified by Glycemic Control Status

The multivariate-adjusted mean OHIP-14T scores and differences in the seven domains among the PT groups stratified by the diabetic control type are listed in Table 4. Patients with well-controlled T2DM in the PT group demonstrated significantly higher scores than did those in the NPD group in two domains of OHIP-14T, namely physical pain (adjusted difference (aDiff) = 0.63, 95% CI = 0.15, 1.10) and psychological discomfort (aDiff = 1.03, 95% CI = 0.40, 1.66). The NPT group had significantly higher scores than the NPD group in four domains of OHIP-14T, namely physical pain (aDiff = 0.70, 95% CI = 0.14, 1.26), psychological discomfort (aDiff = 0.81, 95% CI = 0.06, 1.55), psychological disability (aDiff = 0.49, 95% CI = 0.03, 0.95), and social disability (aDiff = 0.41, 95% CI = 0.03, 0.79). No significant difference was observed for any of the OHIP-14T domains in patients with well-controlled T2DM between the PT and NPT groups. No significant difference was noted for any of the OHIP-14T domains in patients with poorly controlled T2DM between the PT and NPD groups. The NPT group had significantly higher scores than the NPD group in two domains of OHIP-14T, namely physical pain (aDiff = 0.82, 95% CI = 0.38, 1.27) and psychological discomfort (aDiff = 0.97, 95% CI = 0.33, 1.61). The PT group had significantly lower scores than the NPT group in three domains of OHIP-14T, namely physical disability (aDiff = −0.68, 95% CI = −1.23, −0.12), physical pain (aDiff = −0.73, 95% CI = −1.22, −0.23), and social disability (aDiff = −0.38, 95% CI = −0.72, −0.44).

## 4. Discussion

Prior to this research, no literature had explored the fact that periodontal care behaviors may directly affect glycemic control status. We observed a strong relationship between periodontal self-care behavior (i.e., interdental cleaning) and poor glycemic control in T2DM patients. Furthermore, the key finding of this study is that among poorly controlled T2DM patients, the PT group demonstrated a more favorable OHQoL than the NPT group, particularly in physical disability and physical pain domains; however, the OHQoL did not differ significantly in patients with well-controlled T2DM between the PT and NPT groups. Moreover, patients in the PT group were more likely to have periodontal self-care behaviors (i.e., Bass brushing technique and interdental cleaning). 

A strong association between an absence of interdental cleaning and poor glycemic control was found in the present study. The use of interdental cleaning device as an oral hygiene behavior can reduce the amount of dental plaque. A higher frequency of interdental cleaning was associated with lower levels of periodontal disease and caries, as well as lower numbers of missing teeth [29]. Periodontal inflammation negatively affects glycemic control [30], and this might explain why patients who did not have interdental cleaning behaviors had an increased risk of having poorly controlled diabetes in our research. Moreover, the present study found that patients with a higher level of attitude toward periodontal health had a greater likelihood of having poor glycemic control status. Our finding is inconsistent with a systematic review that showed that people with diabetes have poor oral health attitudes [31]. A majority of diabetic patients do not receive information on oral health risks in relation to their diabetes or advice on perinodal health care from diabetes care providers. Instead, diabetic patients may receive oral health counseling by dental health care professionals at their dental visit, which improves their awareness and attitude in related to oral health. 

We found a significant difference in the OHQoL between the PT and NPT groups among patients with poor glycemic control; however, no significant difference was noted in any of the domains of OHIP-14T between the PT and NPD groups. A systematic review indicated that nonsurgical therapy can improve the OHQoL of adults with periodontal disease [32]. Patients with poorly controlled diabetes are more susceptible to periodontal disease [23,24]; they might have relatively poor baseline periodontal status and may consciously feel great improvement in their periodontal status after receiving treatment despite the fact that their actual clinical improvements are not as great as those of patients with well-controlled T2DM.

The most favorable OHQoL was observed in both the glycemic control groups comprised of patients with T2DM but without periodontal disease. Several studies have confirmed that patients with periodontal disease have a poorer OHQoL than those without periodontal disease [14,33]. A similar finding emerged from our study. Therefore, to maintain periodontal health and to enhance OHQoL, daily oral health care is crucial. The American Dental Association lists periodontitis as a complication of diabetes and suggests that patients with diabetes must receive preventive oral health care [34]. Thus, to maintain a healthy periodontal tissue, patients with diabetes should pay attention to their oral health status and follow the treatments recommended by medical staff.

Patients with poorly controlled T2DM in the PT and NPT groups had significant differences in the scores for the physical disability, physical pain, and social disability domains of OHIP-14T. Physical disability refers to the difficulty in eating specific foods and discontinuity in eating during mealtime. Patients with poorly controlled T2DM may develop more severe periodontal disease than those with well-controlled T2DM. Patients with tooth mobility may have dietary restrictions or masticatory disturbances [35] and need to change their eating habits—eating soft foods or changing their cooking methods, for example. Patients with root exposure may experience tooth sensitivity when eating or drinking hot or cold foods. Thus, PT may improve the chewing ability and tooth sensitivity status of these patients. Physical pain refers to the actual pain caused by dental problems; this pain may include soreness of the jaw or other discomfort during mealtime. Our results demonstrated that periodontal disease was correlated with physical pain in patients with T2DM and periodontal disease, corroborating the findings of studies that have not focused on patients with diabetes [14,36]. Though periodontal disease does not directly affect the jaw, we speculate that the pain caused by gingival inflammation may radiate into the jaw area. Severe periodontal disease may cause tooth mobility and even tooth loss, potentially leading to burden when chewing. Social disability refers to a condition of being easily enraged by others or inability to accomplish daily work or activities. Irritability in social situations was the main reason for the difference in scores in this dimension. Patients with dental problems, such as toothache or gingival inflammation, may have diminished emotional self-control. 

Our study found that patients with T2DM who had PT experience were more likely to have periodontal care behaviors, particularly the Bass tooth brushing method and interdental cleaning behavior. The conventional nonsurgical periodontal treatment includes supragingival and subgingival tooth debridement and oral hygiene instruction [37], which might explain why diabetic patients in the PT group had higher probability to use the Bass method and to have interdental cleaning than patients in the NPT group. This finding is consistent with that of a study not focused on people with diabetes in identifying individuals who performed proximal tooth cleaning daily, which was significantly increased during the nonsurgical treatment period [38,39]. We also discovered that gender, education level, and attitude toward periodontal health were correlated with interdental cleaning behavior, and the findings are consistent with those of previous studies [40]. Though the influence of gender on compliance were contradictory, a review that included 22 studies related to the gender factor found that female patients had better compliance in half of the studies [41]. Education level is another important influencing factor for interdental cleaning behavior. A previous study found that educational level had a direct influence on both patient’s knowledge and behavior regarding main oral diseases [42].

This study has several limitations. We were unable to claim a causal relationship in this case–control study. Moreover, PT experience and other variables included in the analyses were based on self-reported information. The measure of the PT status in the self-report survey was not very precise. Social desirability bias is an important concern when interpreting the results because patients may under- or over-report their behaviors. However, some studies have revealed that self-reported data are quite valid and reliable when the participants’ privacy is protected. Finally, this study was conducted in southern Taiwan; therefore, findings should be generalized with caution. The survey was conducted in multiple medical centers; therefore, this limitation potentially had a minimal effect.

## 5. Conclusions

Patients with poorly controlled T2DM who had received PT exhibited a more favorable OHQoL than those who did not receive treatment; this suggested that patients with T2DM and poor glycemic control must develop periodontal care behaviors and receive proper PT, if necessary, to improve their OHQoL. In addition, physicians should actively provide information regarding the maintenance of good periodontal health and self-care behaviors to patients with T2DM. It is recommended that patients with T2DM are motivated to engage in good periodontal self-care behaviors and are provided dental referrals as a routine part of diabetes care.

## Figures and Tables

**Figure 1 ijerph-16-04011-f001:**
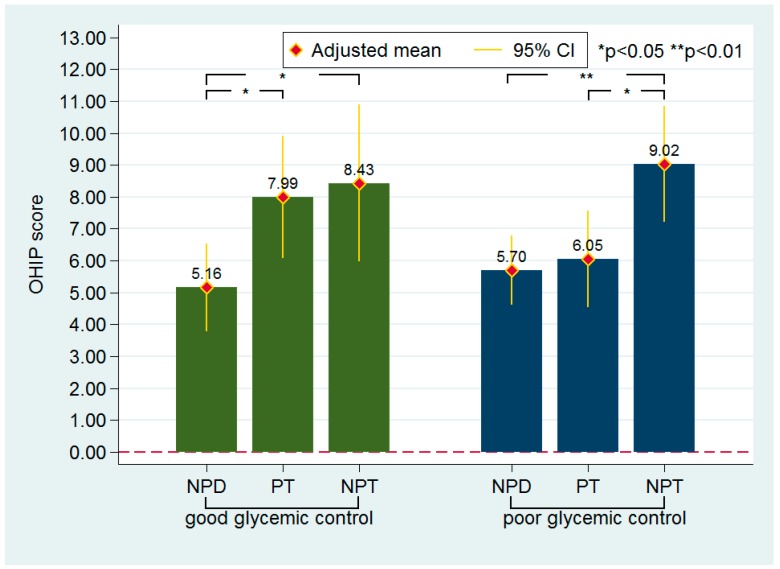
The levels of oral health-related quality of life (OHQoL) in different periodontal treatment experience groups stratified by glycemic control status. NPD: non-periodontal disease; NPT: non-periodontal treatment; PT: periodontal treatment. Adjusted means indicate the estimated values adjusted for the age, gender, type 2 diabetes mellitus duration, education level, and routine dental visit.

**Table 1 ijerph-16-04011-t001:** Distribution of sociodemographic characteristics stratified by glycemic control status in type 2 diabetes mellitus (T2DM) patients.

Factor/Category	Diabetes Control				*p*
	Well-Controlled		Poor-Controlled		
	*N*	%	*N*	%	
Gender					0.854
Male	112	56.6	139	57.4	
Female	86	43.4	103	42.6	
Age (mean ± SD) ^a^	56.9 ± 9.9		56.9 ± 8.9		0.979
Duration of diabetes (year) ^b^					<0.001
<5	121	64.0	112	47.3	
5–10	45	23.8	53	22.4	
>10	23	12.2	72	30.4	
Education level					0.240
Less than junior high school	98	49.5	139	57.4	
High school	65	32.8	69	28.5	
Higher than technical school/college	35	17.7	34	14.1	
Yearly household income (NTD) ^b^					0.776
<450,000	95	50.0	124	52.8	
450,000–600,000	37	19.5	40	17.0	
>600,000	58	30.5	71	30.2	
Drinking					0.096
Yes	13	6.6	27	11.2	
No	185	93.4	215	88.8	
Betel nut chewing					0.032
Yes	4	2.0	15	6.2	
No	194	98.0	227	93.8	
Smoking					0.044
Yes	23	11.6	45	18.6	
No	175	88.4	197	81.4	
Perceived oral health					0.248
Very good/Good	55	27.8	52	21.5	
Common	89	45.0	125	51.7	
Very poor/Poor	54	27.3	65	26.9	
Periodontal treatment experience					0.755
NPD	109	55.1	128	52.9	
PT	56	28.3	67	27.7	
NPT	33	16.7	47	19.4	

NPD: non-periodontal disease; PT: periodontal treatment; NPT: non-periodontal treatment. ^a^ Two sample t-test ^b^ Missing values represent refusal to answer or forgetting the answer.EG: experimental group. CG: control group.

**Table 2 ijerph-16-04011-t002:** The adjusted odds ratio (aOR) for glycemic control status associated with selected variables in T2DM patients.

Variable/Category	Glycemic Control					
	Poor-Controlled		Well-Controlled			
	N	%	N	%	aOR *	(95%CI)
Gender						
Female (ref.)	103	42.6	86	43.4	1.00	
Male	139	57.4	112	56.6	0.94	(0.61,1.43)
Age, mean ± SD	56.9 ± 8.9		56.9 ± 9.9		0.99	(0.97,1.02)
Education level						
Higher than technical school/college (ref.)	34	14.1	35	17.7	1.00	
High school	69	28.5	65	32.8	1.12	(0.60,2.08)
Less than junior high school	139	57.4	98	49.5	1.60	(0.85,3.00)
Betel nut chewing						
No (ref.)	227	93.8	194	98.0	1.00	
Yes	15	6.2	4	2.0	2.77	(0.82,9.34)
Smoking						
No (ref.)	197	81.4	175	88.4	1.00	
Yes	45	18.6	23	11.6	1.50	(0.81,2.78)
Oral health knowledge	3.4 ± 2.1		3.3 ± 2.2		0.99	(0.89,1.10)
Attitude toward periodontal health	27.0 ± 2.6		26.6 ± 2.9		1.11	(1.02,1.21)
Periodontal treatment experience						
NPD (ref.)	128	52.9	109	55.1	1.00	
PT	67	27.7	56	28.3	1.13	(0.70,1.81)
NPT	47	19.4	33	16.7	1.28	(0.74,2.20)
Tooth-brushing time						
≥2 mins (ref.)	92	38.5	67	33.8	1.00	
<2 mins	147	61.5	131	66.2	0.75	(0.50,1.13)
Tooth-brushing method						
Bass brushing technique (ref.)	70	29.2	70	35.5	1.00	
Others	170	70.8	127	64.5	1.32	(0.86,2.03)
Interdental cleaning						
Yes (ref.)	38	15.7	47	23.7	1.00	
No	204	84.3	151	76.3	1.76	(1.05,2.96)
Routine dental visit						
Yes (ref.)	34	14.1	27	13.6	1.00	
No	208	86.0	171	86.4	0.86	(0.48,1.53)

NPD: non-periodontal disease; PT: periodontal treatment; NPT: non-periodontal treatment. * Adjusted values are based on multivariate logistic regression model.

**Table 3 ijerph-16-04011-t003:** Periodontal treatment experience and selected variables related to periodontal care behaviors in T2DM patients in multivariate logistic regression model

Variable/Category	Periodontal Care Behaviors							
	Tooth-Brushing Time ≥2 min vs. <2 min		Tooth-Brushing Method Bass Method vs. Others		Interdental Cleaning Yes vs. No		Routine Dental Visit Yes vs. No	
	aOR	(95% CI)	aOR	(95% CI)	aOR	(95% CI)	aOR	(95% CI)
Gender								
Male (ref.)	1.00		1.00		1.00		1.00	
Female	1.17	(0.77,1.77)	0.99	(0.64,1.52)	1.82	(1.07,3.10)	0.94	(0.52,1.71)
Age	1.01	(0.99,1.04)	0.97	(0.95,0.99)	1.00	(0.97,1.03)	0.99	(0.52,1.71)
Education level								
Less than junior high school (ref.)	1.00		1.00		1.00		1.00	
High school	1.06	(0.64,1.75)	1.23	(0.74,2.05)	1.92	(1.03,3.60)	1.40	(0.69,2.82)
Higher than technical school/college	2.00	(1.08,3.70)	1.05	(0.55,2.00)	2.95	(1.42,6.11)	2.17	(0.99,4.76)
HbA1c(%)	1.04	(0.95,1.14)	0.99	(0.89,1.10)	0.83	(0.68,1.00)	0.88	(0.72,1.08)
Oral health knowledge	1.01	(0.91,1.13)	0.91	(0.82,1.03)	1.03	(0.90,1.19)	1.02	(0.87,1.18)
Attitude toward periodontal health	0.99	(0.91,1.07)	1.09	(1.00,1.19)	1.15	(1.04,1.27)	1.10	(0.99,1.22)
Periodontal treatment experience								
NPD (ref.)	1.00		1.00		1.00		1.00	
PT	1.58	(0.99,2.52)	1.77	(1.09,2.87)	2.47	(1.39,7.39)	1.84	(0.96,3.51)
NPT	1.34	(0.78,2.30)	1.46	(0.83,2.58)	1.41	(0.70,2.84)	1.85	(0.88,3.89)

NPD: non-periodontal disease; PT: periodontal treatment; NPT: non-periodontal treatment.

**Table 4 ijerph-16-04011-t004:** Multivariate-adjusted means and differences in scores in the seven Oral Health Impact Profile-14 domains in periodontal treatment groups stratified by glycemic control status.

	Well−Controlled							Poor−Controlled						
	NPD	PT vs. NPD		NPT vs. NPD		PT vs. NPT		NPD	PT vs. NPD		NPT vs. NPD		PT vs. NPT	
	Mean	Diff	(95% CI)	Diff	(95% CI)	Diff	(95% CI)	Mean	Diff	(95% CI)	Diff	(95% CI)	Diff	(95% CI)
Functional limitation	0.62	0.38	(−0.09,0.85)	0.52	(−0.03,1.08)	−0.15	(−0.76,0.47)	0.57	0.33	(−0.07,0.73)	0.43	(−0.02,0.88)	−0.10	(−0.61,0.41)
Physical disability	1.06	0.54	(−0.02,1.10)	0.35	(−0.31,1.01)	0.19	(−0.54,0.92)	1.27	−0.21	(−0.65,0.23)	0.47	(−0.03,0.97)	−0.68	(−1.23, −0.12)
Physical pain	0.54	0.63	(0.15,1.10)	0.70	(0.14,1.26)	−0.07	(−0.69,0.55)	0.65	0.10	(−0.29,0.49)	0.82	(0.38,1.27)	−0.73	(−1.22,−0.23)
Psychological discomfort	1.70	1.03	(0.40,1.66)	0.81	(0.06,1.55)	0.22	(−0.60,1.05)	1.95	0.29	(−0.27,0.85)	0.97	(0.33,1.61)	−0.68	(−1.39,0.04)
Psychological disability	0.51	0.03	(−0.36,0.42)	0.49	(0.03,0.95)	−0.46	(−0.97,0.05)	0.54	0.04	(−0.31,0.39)	0.39	(−0.01,0.79)	−0.35	(−0.80,0.10)
Social disability	0.32	0.05	(−0.27,0.37)	0.41	(0.03,0.79)	−0.36	(−0.78,0.06)	0.34	−0.15	(−0.41,0.12)	0.23	(−0.07,0.54)	−0.38	(−0.72,−0.44)
Handicap	0.41	0.17	(−0.19,0.54)	−0.01	(−0.44,0.42)	0.18	(−0.29,0.66)	0.37	−0.06	(−0.35,0.24)	0.01	(−0.32,0.35)	−0.07	(−0.44,0.31)

NPD: non-periodontal disease; PT: periodontal treatment; NPT: non-periodontal treatment. Means and differences were adjusted for age, gender, type 2 diabetes mellitus duration, education level and routine dental visit.

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
