# Peer review of "Periodontal Treatment Experience Associated with Oral Health-Related Quality of Life in Patients with Poor Glycemic Control in Type 2 Diabetes: A Case-Control Study"

_ijerph, 2019, doi:10.3390/ijerph16204011_

Round 1
Reviewer 1 Report
Although the article does not bring any novel findings, the paper is understandable, design of the study is good. Results are described properly and discussion meets the outcome of the study. Conclusions highlight the need of inter-disciplinary approach to this health issue.
I agree with publication.
Author Response
#Reviewer 1:
Thanks for the constructive comments
Although the article does not bring any novel findings, the paper is understandable, design of the study is good. Results are described properly and discussion meets the outcome of the study. Conclusions highlight the need of inter-disciplinary approach to this health issue. I agree with publication.RE: Thank you.
Reviewer 2 Report
please revise the paper for grammar and style, there are many typos
deficient literature review, update
add the significant statistical findings to the abstract, what was the most important clinical message
what was your null hypothesis? did you reject/accept/partially reject it?
provide information on sample size calculation, questionnaire, or any index or variable you assessed/measured, provide tables showing components of the indices in separate subheadings
short discussion, develop, update
Author Response
#Reviewer 2:
Thanks for the constructive comments
Please revise the paper for grammar and style, there are many typosRE: Before 1st version submission, we had sent to the English Editing Company for grammar check (attached please find the certificate of English editing). The style was revised based on IJERPH.
deficient literature review, update
RE: The literatures review was revised and updated in introduction. (Line 44-97)
add the significant statistical findings to the abstract, what was the most important clinical message
RE: The significant statistical findings were added in the abstract (line 36-39).We have added the most important clinical message, which is “diabetic patients with poor glycemic control must improve periodontal care practices and receive proper PT, if necessary, to improve their OHQoL” (Line 39-40).
what was your null hypothesis? did you reject/accept/partially reject it?
RE: Our study aims were revised(Line 94-97).
We tested two hypotheses:
The periodontal treatment (PT) experience and periodontal care behaviors are associated with poor glycemic control. There is a significant difference in OHQoL between patients with poorly controlled T2DM who had received PT and those who did not receive PT.Our findings were:
Periodontal care behavior (i.e. interdental cleaning) was significantly more likely to be associated with poor glycemic control in patients with T2DM (Accept null hypothesis). However, periodontal treatment was not significantly associated with glycemic control (Reject null hypothesis). The PT group had significantly lower level of OHQoL than the Non-PT group among patients with poorly controlled T2DM(Accept null hypothesis).provide information on sample size calculation, questionnaire, or any index or variable you assessed/measured
RE: We added the information on sample size calculation (Lines 116-118). Information regarding questionnaire and variables, please see
“2.3. Questionnaire development and measures” (Lines 120-174).
provide tables showing components of the indices in separate subheadings
RE: Subheadings for tables/figure were provided (Lines 189-243).
short discussion, develop, update
RE: We have developed an updated the discussion based on reviewer’s suggestions (Lines 265-352).
Reviewer 3 Report
Dear Authors,
Congrats for the present study entitled “Periodontal Treatment Experience Associated with 2 Oral Health-Related Quality of Life in Patients with 3 Poor Glycemic Control in Type 2 Diabetes: A 4 Case-Control Study”, that was a well-conducted study.
I have only one suggestion:
*Materials and Methods:
Please, start your Materials and Methods with the “2.5. Human ethics”.
Thank you for sharing your research.
Best regards,
Reviewer#
Author Response
#Reviewer 3:
Thanks for the constructive comments
Congrats for the present study entitled “Periodontal Treatment Experience Associated with 2 Oral Health-Related Quality of Life in Patients with 3 Poor Glycemic Control in Type 2 Diabetes: A 4 Case-Control Study”, that was a well-conducted study.RE: Thank you.
I have only one suggestion:
*Materials and Methods:
Please, start your Materials and Methods with the “2.5. Human ethics”.
RE: We have revised. Please see 2.1. Human ethics (Line 101-104).
Reviewer 4 Report
[Suggestions]
A very interesting paper! In particular, Figure 1 and L. 202-204; "Patients with poorly controlled T2DM in the PT group had a significantly lower OHIP-14T score than did those in the NPT group (6.05 vs. 9.02)."
But, just note!
The authors should check the data in the abstract (L. 36-38); "This group had significantly lower OHIP-14 scores than the NPT group (5.93 vs. 8.99), ......" Maybe a typographical error!!
Abstract (L. 25-27) and Introduction (L. 71-72):
The referee would rather say that the manuscript, in the Abstract and Introduction, needs to express rationales of this study in more detail, such as, Why did the authors want to know the association among (i.e., want to examine) periodontal treatment experience, periodontal care behaviors, and oral health-related quality of life in patients with type 2 diabetes mellitus? and Why did the authors focus on patients with poor glycemic control?
Author Response
#Reviewer 4:
Thanks for the constructive comments
A very interesting paper! In particular, Figure 1 and L. 202-204; "Patients with poorly controlled T2DM in the PT group had a significantly lower OHIP-14T score than did those in the NPT group (6.05 vs. 9.02)."But, just note!
RE: Thank you.
The authors should check the data in the abstract (L. 36-38); "This group had significantly lower OHIP-14 scores than the NPT group (5.93 vs. 8.99), ......" Maybe a typographical error!!RE: We have revised (Line 36).
Abstract (L. 25-27) and Introduction (L. 71-72):
The referee would rather say that the manuscript, in the Abstract and Introduction, needs to express rationales of this study in more detail, such as, Why did the authors want to know the association among (i.e., want to examine) periodontal treatment experience, periodontal care behaviors, and oral health-related quality of life in patients with type 2 diabetes mellitus? and Why did the authors focus on patients with poor glycemic control?
RE: Rationales of this study were revised in the abstract (Line 25-26) and introduction (Line 78-97).